# Optimization of Two-Phase Ejector Mixing Chamber Length under Varied Liquid Volume Fraction

**DOI:** 10.3390/e25010007

**Published:** 2022-12-21

**Authors:** Jia Yan, Yuetong Shu, Jing Jiang, Huaqin Wen

**Affiliations:** 1School of Civil Engineering and Architecture, Southwest University of Science and Technology, Mianyang 621010, China; 2School of Artificial Intelligence, Yantai Institute of Technology, Yantai 264003, China

**Keywords:** ejector, entrainment ratio, liquid volume fraction, numerical simulation, mixing chamber length

## Abstract

The ejector performance varies with the mixing chamber length which is largely dependent on the fluid liquid volume fraction at the inlet. In this study, numerical simulations are conducted to optimize two mixing chamber lengths of a two-phase ejector under varied liquid volume fractions of 0–0.1 in two inlet fluids. The main findings are as follows: (1) The two optimal lengths of constant-pressure and constant-area mixing chambers are identified within 23–44 mm and 15–18 mm, respectively, when the primary inlet fluid is in two-phase; (2) the two optimal lengths are 2–5 mm and 9–15 mm, respectively, when the secondary inlet fluid is in two-phase; (3) when both inlets are in two-phase, the two optimal lengths are ranged in 5–23 mm and 6–18 mm; (4) little liquid within inlet fluid has a significant influence on ejector performances; and (5) optimal constant-pressure mixing chamber length and the sum of the two optimal lengths increase with the primary flow inlet liquid volume fraction but decrease with that of the secondary flow inlet.

## 1. Introduction

With the rapid development of technology, energy consumption has restricted economic and social development. Therefore, it is quite urgent to improve energy efficiency in air-conditioning and refrigeration devices [1,2]. Various studies have been conducted to improve the performance of refrigeration systems and thus solve environmental issues [3,4,5]. Moreover, the performance of refrigeration systems has been enhanced by adopting advanced technologies [6,7,8,9]. For the refrigeration needed in refrigerated trucks that always need air-conditioning services and refrigerating or freezing purposes for food storage, a pressure regulating valve (PRV) is equipped between two evaporators to keep the required pressure difference [10,11], which causes many irreversible throttling losses. Therefore, an ejector is used to replace the PRV and partially recover the throttling losses [12,13,14]. The schematic of a typical simplified EMERS with two temperature levels is shown in Figure 1 [15]. An EMERS has some advantages such as low operating costs [16,17].

When the EMERS is used in refrigerated trucks, the essential device of the system is the ejector [18,19]. The two flows of the refrigerant flows mix in the ejector and enter the compressor with a pressure lift [20]. By optimizing the area ratio (AR) and the nozzle exit position (NXP) and so on, ejector performance can be improved [21,22].

High entrainment performance of the ejector can be achieved if the two flows are mixed well [23]. Nakagawa et al. [24] studied the effect of mixing the length of a transcritical CO_2_ two-phase ejector with a rectangular cross-section, and they claimed that the 15 mm of mixing length can produce good ejector performance. Sarkar et al. [25] showed that the constant-area mixing chamber cross-section area affects ejector performance mainly depending on the ejector inlet conditions. Banasiak et al. [26] also proved that the ejector performance largely depends on the mixing chamber length in a small ejector-based R744 transcritical heat pump system. Jeon et al. [27] studied an ejector mixing length and improved system performance. Fu et al. [28] optimized the mixing chamber throat diameter to improve the steam ejector performance. By using three-dimensional numerical simulations, Dong et al. [29] studied the effects of the mixing chamber length, and the best ejector performance was obtained within a certain range of the mixing chamber length.

In many cases, the ejector operates with a gas-liquid mixture of primary flow or gas-liquid mixture of secondary flow. Hemidi et al. [30] conducted the study when the water droplets were ejected into primary air flow, which improved off-design performance. Yuan et al. [31] investigated a two-phase ejector experimentally and numerically. Similarly, according to the study of Yuan et al. [31], Chen et al. [32] also studied the two-phase secondary flow ejector performance. They claimed that ER and PRR would decrease when the induced flow is accompanied by water. Aliabadi et al. [33] investigated the effects of primary nozzle inlet wetness in the range of 0–1%. Their results indicate that the water droplets make an ER improvement.

To the best of the authors’ knowledge, there is no study on the effects of mixing chamber length under different liquid volume fractions (LVF) which means the liquid volume percentage in the two-phase flow on ejector performance used refrigerant of C_2_H_2_F_4_ as displayed in Figure 1. With our former study [15], it was known that when the LVF of the two inlet flows varies, it may have an undesirable effect on ejector performance, and thus, the ejector with original geometries may be in malfunction.

Thus, this study aims to optimize the constant-pressure mixing section length (L_pm_) and constant-area mixing section length (L_am_) of a two-phase ejector under different primary and secondary flow liquid volume fractions. The details of the work in this paper are:to identify optimal L_pm_ under varied secondary flow liquid volume fraction;to find the optimal L_pm_ under varied primary flow liquid volume fraction;with optimal L_pm_, to search for the optimal L_am_ under varied secondary flow liquid volume fraction;with optimal L_pm_, to optimize the L_am_ under varied primary flow liquid volume fraction.

## 2. CFD Modeling and Validation

The schematic of the ejector is presented in Figure 2 [15]. Its initial geometrical parameters are presented in Table 1, and boundary conditions are presented in Table 2.

To simulate the complex flow regime, the governing equations are the steady Reynolds Averaged Navier-Stokes equations [34,35].

Fluent 19.0 is used for the simulation. According to Palacz et al. [36], the differences in the results for the 3-D and 2-D models are negligible; therefore, the axisymmetric two-dimensional model is utilized in this CFD simulation. The properties of the working fluid are derived from NIST. Besagni et al. [37], Exposito-Carrillo et al. [38], and Croquer et al. [39] found that the k-omega SST model generally performed better in simulating the single-phase ejector; however, the realizable k-epsilon model is employed for two-phase ejector [35]. Meanwhile, near-wall refinement is used in the regions where large pressure and temperature gradient are possible to better capture shock waves and complex internal flow details. In addition, sensitivity analysis on wall treatments is performed and the first grid locates at 30 < y + < 300, which gives accurate results [15].

The PRESTO algorithm is applied to pressure-solving. Moreover, the second-order upwind discretization scheme is employed for density, momentum, energy, turbulent kinetic energy, and turbulent dissipation rate solving. All equations are iterated until the residuals are below 10–6. In addition, for an optimization study, consistent convergence of the CFD solution is sometimes difficult, especially when the solution is likely to be quasi-steady-state due to turbulence and the multi-phases. The convergence can often be slow, and the residual can remain stagnating or oscillating above-chosen convergence criteria.

Figure 3 presents the 2-D axisymmetric quadrilateral grid configuration for the baseline ejector. As shown in Figure 3, the pressure and velocity at Point A and Point B are used to detect the influence of the cell number. Table 3 and Table 4 display the grid independence verification results. Pressure and velocity errors with area A and area B are less than 0.5%, indicating that the results are within the acceptable ranges; thus, the medium one with a grid number of 83,100 is selected.

The CFD model is validated by the void fraction inside the ejector which is based on the experimental results [15]. Take a typical case as an example, when LVF_1_ is 0.1 and LVF_2_ is 0. As shown in Figure 4, the maximum discrepancy is within 7.9%. The maximum deviation of α for many other ejector dimensions does not exceed 15%; thus, the model can be used in the following simulation.

As for the selection of a convergent nozzle, the comparison between the converging and the converging-diverging nozzle is presented below. For single-phase primary and secondary flow, based on the initial geometries, when the NXP is fixed at 0 mm and primary nozzle diverging section length is varied with 0 mm, 2 mm, 4 mm, and 6 mm, mass flow rates and ER are displayed in Figure 5. It is clear that, with the increase of the divergent section length, m_1_ has little change, m_2_ decreases significantly, and correspondingly, ER decreases with the increases in divergent section length. Moreover, the corresponding Mach number contours are shown in Figure 6. Hence, the converging nozzle is selected.

## 3. Results and Discussion

### 3.1. Optimization of L_pm_

#### 3.1.1. Effect of Two-Phase Primary Flow

The L_pm_ is a key design parameter because it determines the mixing efficiency which indicates the ejector performance. Several groups of the L_pm_ are selected for analysis of the ejector performance under different LVFs of primary flow.

Figure 7 displays the ER with L_pm_ under LVF_2_ = 0 and varied LVF_1_ (the unfilled point indicates the baseline ejector model). It can be observed that for different LVF_1_, ER always first rises moderately and then decreases steeply. To be specific, for LVF_1_ of 0.02, when L_pm_ is increased from 8 mm to 50 mm, ER increases and peaks at L_pm_ of 23 mm which is magnified for the legend. After the peak value, ER drops slowly at first, then it falls suddenly, and even backflow occurs. Similarly, ER under LVF_1_ of 0.04 also rises first and then reaches the peak of 0.237 at L_pm_ = 38 mm; the maximum ER increases by 26.58% over the baseline ejector. When LVF_1_ are 0.06, 0.08, and 0.1, all the highest ER (0.145, 0.0922, and 0.0596, respectively) are achieved with L_pm_ of 44 mm. In addition, after the maximum value, the ER decreases suddenly to a negative value. To elucidate the abrupt drop of ER, contours of static pressure and axial static pressure distribution for the L_pm_ of 44 mm, 47 mm, and 50 mm are displayed in Figure 8a,b, respectively. Note that the amount of secondary fluid mass flow depends on how much pressure drop is induced at the outlet of the primary nozzle. Obviously, for L_pm_ = 44 mm, the static pressure rises smoothly, which can improve ejector performance. While for L_pm_ of 47 mm and 50 mm, higher pressure is generated, which weakens the ejector performance. When L_pm_ increases to 50 mm, the area of the high-pressure region increases, which further results in a decrease in ER. In addition, from another perspective, with the increase of L_pm_, these two fluids mix more sufficiently, and correspondingly, the entrainment performance is enhanced; however, the increase of L_pm_ will also lead to the increase of frictional loss. Therefore, ejector performance suddenly decreases as L_pm_ increases to a certain value. As displayed by the velocity contours in Figure 8c, the energy loss increases largely when L_pm_ increases from 44 mm to 47 mm. For this purpose, L_pm_ should not exceed 44 mm for a proper operation of the ejector.

In addition, compared with the baseline ejector model with an L_pm_ of 11 mm, the ejector under two-phase primary flow operation has a much longer optimal L_pm_. That is to say, the optimal L_pm_ seriously deviates from the baseline ejector model, but when L_pm_ is more than 44 mm, the performance of the ejector drops drastically, which should be avoided.

#### 3.1.2. Effect of Two-Phase Secondary Flow

Optimization of L_pm_ under different LVF_2_ and fixed LVF_1_ of 0 is conducted in this section. Figure 9 portrays the change trends of ER with L_pm_. To be specific, when LVF_2_ is 0.02, ER first increases and then peaks at 1.99 when L_pm_ equals 5 mm. That is to say, for LVF_2_ of 0.02, there exists an optimal L_pm_ of 5 mm, which is less than the L_pm_ of the baseline ejector of 11 mm. When LVF_2_ varies from 0.04 to 0.1, all ERs rise first and then decrease with an increase in L_pm_. Moreover, as displayed by the magnified point in Figure 9, the optimal L_pm_ are the same and all equal 2 mm, which is less than the L_pm_ of the baseline ejector as well. The maximum ERs are 2.36, 2.62, 2.78, and 2.91 for LVF_2_ of 0.04, 0.06, 0.08, and 0.1, respectively, or the maximum ER increases with increasing LVF_2_. Therefore, a higher ER is generated with a shorter L_pm_ under a two-phase secondary flow. Furthermore, it can be found that the optimal L_pm_ of a two-phase secondary flow is much shorter than the optimal L_pm_ of a two-phase primary flow.

With the results of Section 3.1.1 and Section 3.1.2, the optimal L_pm_ is in the range of 23–44 mm when LVF_1_ = 0.02~0.1 and LVF_2_ = 0, and the longest optimal L_pm_ of 44 mm is obtained at LVF_1_ = 0.1 and LVF_2_ = 0. Moreover, the optimal L_pm_ is in the range of 2–5 mm when LVF_2_ = 0.02~0.1 and LVF_1_ = 0, and it can be said that the shortest optimal L_pm_ of 2 mm is obtained at LVF_1_ = 0 and LVF_2_ = 0.1. That is, when the LVF of both inlets are very different, the optimal L_pm_ also has a striking difference.

#### 3.1.3. Effect of Two-Phase Primary and Secondary Flows

The above two sections are carried out under the circumstance that one of the ejector inlets does not contain liquid. Optimization of L_pm_ when both the primary and secondary flows contain liquid, by varying the LVF_1_ and LVF_2_, respectively, relevant simulation results are given below.

(a).Varied LVF_2_ with fixed LVF_1_

Figure 10 depicts the relationship between L_pm_ and ER under LVF_1_ = 0.02 and LVF_2_ = 0.02~0.1. It is readily found that for diversified LVF_2_, ER always initially increases and then decreases along with the increase of L_pm_. When LVF_2_ changes from 0.04 to 0.1, the optimal L_pm_ are all 5 mm, which is less than the L_pm_ of the baseline ejector. The highest values of ER for LVF_2_ from 0.04 to 0.1 are 1.65, 1.91, 2.07, and 2.2, respectively. In addition, in comparison with the baseline ejector, the corresponding maximum ERs increase by 6.64%, 13.58%, 16.81%, and 20.2%, respectively.

Figure 11 is the ER with L_pm_ under fixed LVF_1_ of 0.06 and various LVF_2_. For different LVF_2_, ER always rises first and then consistently reduces along with the increase of L_pm_. To be specific, for LVF_2_ of 0.02, when L_pm_ increases from 2 mm to 20 mm, ER increases and arrives at its peak value of 0.68 at the L_pm_ of 17 mm. Furthermore, the maximum ER increases by 1.54% over the baseline ejector. Similarly, the changing trend of ER for LVF_2_ = 0.04 is basically the same as that of LVF_2_ = 0.02. The difference is that the optimal L_pm_ for LVF_2_ = 0.04 is 11 mm, which is less than the L_pm_ of LVF_2_ = 0.02. For LVF_2_ of 0.06 and 0.08, both the peak values of ERs are obtained at the L_pm_ of 8 mm, which is less than the L_pm_ of the baseline ejector. Moreover, the maximum ER increases by 1.24% and 3.34%, respectively. As for the LVF_2_ of 0.1, the maximum ER is 1.66 with the L_pm_ = 5 mm, and the maximum ER increases by 6.98%. It is worth mentioning that when L_pm_ deviates from the optimal value, the performance of the ejector will be greatly reduced. Furthermore, it is obviously observed that when LVF_2_ increases, the optimal L_pm_ gets smaller. Compared with Figure 10, it can also be found that when LVF_1_ increases from 0.02 to 0.06, for each LVF_2_, the optimal L_pm_ increases a little.

Figure 12 illustrates the changing trend of ER with L_pm_ under fixed LVF_1_ of 0.1 and various LVF_2_. For LVF_2_ of 0.02, the maximum ER is 0.423 with L_pm_ = 23 mm, which increases by 9.59%. For LVF_2_ of 0.04, the highest value of ER, 0.744, is achieved at the L_pm_ of 14 mm. Both for LVF_2_ of 0.06 and 0.08, the maximum ERs (0.983 and 1.166, respectively) are obtained when L_pm_ reaches 11 mm. Moreover, similar to Figure 10, as LVF_2_ increases from 0.02 to 0.1, the optimal L_pm_ is reduced gradually, since the optimal L_pm_ is 23 mm, 14 mm, 11 mm, 11 mm, and 8 mm, respectively. However, compared with Figure 10, namely when LVF_1_ increases from 0.06 to 0.1, for each fixed LVF_2_, each optimal L_pm_ increases slightly, but the maximum ER drops.

In general, with the results of Figure 10, Figure 11 and Figure 12, it can be concluded that: (1) for each fixed LVF_1_, when LVF_2_ increases from 0.02 to 0.1, the optimal L_pm_ decreases to different degrees; (2) the optimal L_pm_ is in 5–23 mm; (3) with the increases of LVF_1_, the optimal L_pm_ generally increases; (4) combined with the operating condition of two-phase primary flow (LVF_2_ = 0.02~0.1) as presented in Section 3.1.2, the optimal L_pm_ and ER decrease with an increase of LVF_1_.

(b).Varied LVF_1_ with fixed LVF_2_

Figure 13 is the ER with L_pm_ under fixed LVF_2_ of 0.02 and various LVF_1_. Specifically speaking, for LVF_1_ of 0.02, the ER reaches the peak value of 1.248 at L_pm_ = 8 mm. The maximum ER increases by 0.61% compared with the baseline ejector. In terms of LVF_1_ = 0.04, the ER increases slightly from 0.844 to 0.907, after the highest value, ER drops gradually, and the optimal L_pm_ is 14 mm in this condition. When LVF_1_ is in the range of 0.06 to 0.1, as L_pm_ increases, the increments of the ER do not change a lot. The optimal L_pm_ are 17 mm, 20 mm, and 23 mm for LVF_1_ of 0.06, 0.08, and 0.1, respectively. The corresponding maximum ER increases by 1.54%, 3.34%, and 6.65%, respectively. Generally speaking, for LVF_2_ of 0.02, as LVF_1_ varies from 0.02 to 0.1, the optimal L_pm_, as displayed by the magnified point in Figure 13, becomes larger and larger. 

Figure 14 depicts the effect of L_pm_ on the ER under fixed LVF_2_ of 0.06 and various LVF_1_. ERs always increase initially and then decrease. To be specific, for LVF_1_ of 0.02 and 0.04, both the ERs obtain the maximum value (1.915 and 1.544, respectively) at the L_pm_ of 5 mm, and the maximum ER increases by 13.58% and 4.17%, respectively. When LVF_1_ changes from 0.08 to 0.1, ER increases along with the increase of L_pm_ and obtain the maximum of 1.12 and 0.98, respectively, both the optimum L_pm_ are 11 mm. Overall, when LVF_1_ changes from 0.02 to 0.1, the optimal L_pm_ increases gradually, but all the optimal L_pm_ are no more than the L_pm_ of the baseline ejector. Compared with LVF_1_ = 0.02 as displayed in Figure 13, when LVF_2_ is 0.06, for each LVF_1_, all the maximum ERs increase, but optimal L_pm_ decrease.

Figure 15 displays the impact of L_pm_ on the ER under LVF_2_ = 0.1 and various LVF_1_. The results reveal that all the Ers follow a similar pattern, namely, they increase first and then decrease with the growth of L_pm_. For LVF_1_ varied from 0.02 to 0.06, the maximum Ers (2.2, 1.9, and 1.66, respectively) are all achieved at the L_pm_ of 5 mm, which is slightly less than the baseline ejector. The maximum ER has an increase of 20.2%, 13.68%, and 6.98%, respectively. Moreover, when LVF_1_ increases from 0.08 to 0.1, the optimal L_pm_ increases to 8 mm. The corresponding maximum ERs are 1.47 and 1.32 for LVF_1_ of 0.08 and 0.1, respectively. In addition, the maximum ER increases by 3.01% and 1.4%, respectively. It is noteworthy that for fixed LVF_2_ of 0.1 and various LVF_1_, all the optimal L_pm_ are less than 11 mm. Compared with Figure 14 in which LVF_2_ is 0.06, for each LVF_1_, the maximum ER increases. Moreover, for LVF_1_ = 0.06–0.1, the optimal L_pm_ also increases.

Overall, from Figure 13, Figure 14 and Figure 15, it can be inferred that: (1) for each fixed LVF_2_, the optimal L_pm_ increases with the growth of LVF_1_; (2) with the increase of LVF_2_, the optimal L_pm_ is generally reduced; (3) when both LVF_1_ and LVF_2_ are in the range of 0.02–0.1, the optimal L_pm_ is in the range of 5–23 mm; (4) and, combined with Figure 6 in which LVF_2_ is 0, it can also be concluded that the when LVF_1_ is fixed in the range of 0–0.1, the optimal L_pm_ improves with the growth of LVF_1_, and the ER rises with the rise of LVF_2_.

### 3.2. Optimization of L_am_

Based on the optimal L_pm_ determined in Section 3.1, the following simulations are performed to seek the optimal L_am_.

#### 3.2.1. Effect of Two-Phase Primary Flow

Figure 16 reveals the influence of L_am_ on ER under various LVF_1_ (LVF_2_ = 0). For LVF_1_ of 0.02 and 0.04, when L_am_ increases from 9 mm to 24 mm, the influence of L_am_ is not evident. The ER for LVF_1_ = 0.02 increases from 0.438 at the L_am_ of 9 mm to the maximum of 0.442 at the L_am_ of 18 mm and then drops. For LVF_1_ = 0.04, the ER rises first and reaches the maximum of 0.24 at L_am_ = 15 mm and then drops gradually. The optimal L_am_ for LVF_1_ of 0.02 and 0.04 are 18 mm and 15 mm, respectively. For LVF_1_ of 0.06 and 0.08, the optimal L_am_ are both 15 mm. Nonetheless, when L_am_ exceeds 15 mm, ER decreases abruptly. For LVF_1_ of 0.1, the peak value of ER is 0.064 at the L_am_ of 18 mm, which means the LVF_1_ = 0.1 has a more evident effect on the ER, the reason is that the liquid density is much higher than the vapor density. Likewise, after the peak value, ER drops suddenly. To avoid the malfunction of the ejector, the L_am_ should not exceed 15 mm, and the primary flow should not contain liquid. To identify the cause for the abrupt decrease, contours of static pressure, axial static pressure distribution, velocity contours, and the velocity vector field are displayed in Figure 17a–d, respectively. It can be observed from Figure 17a,b that the static pressure monotonically increases in the mixing chamber when L_am_ is 15 mm and 18 mm. In addition, when L_am_ rises to 21 mm, the mixed fluids have a momentum drop since the increase of the resistance weakens the ejector performance. In other words, when the friction loss caused by the increase of L_am_ exceeds the performance enhanced by the effect of more sufficient mixing, the ejector performance will decrease. Moreover, reflux occurs as illustrated in Figure 17d.

#### 3.2.2. Effect of Two-Phase Secondary Flow

Figure 18 displays the effect of L_am_ on ER under fixed LVF_1_ of 0 and various LVF_2_. Obviously, under all the LVF_2_, ER increases first and then decreases. Nevertheless, for various LVF_2_, the optimal L_am_ is not always the same. Specifically, for LVF_2_ = 0.02, the maximum ER of 2.02 peaks at the L_am_ of 12 mm. For LVF_2_ = 0.04, the optimal L_am_ of 15 mm is the same as that of the baseline ejector. When LVF_2_ is in the range of 0.06–0.1, the optimal L_am_ is at 9 mm.

#### 3.2.3. Effect of Two-Phase Primary and Secondary Flows

All the L_am_ are optimized under LVF_1_ = 0.02~0.1 and LVF_2_ = 0.02~0.1 with an interval of 0.02. Considering the limited space, only two cases (LVF_1_ = 0.1 and LVF_2_ = 0.02~0.1, LVF_2_ = 0.1 and LVF_1_ = 0.02~0.1) are displayed here. 

Figure 19 displays the effect of L_am_ on ER under fixed LVF_1_ of 0.1 and various LVF_2_ (0.02–0.1). When L_am_ increases from 6 mm to 21 mm, for a fixed LVF_2_, the optimal L_am_ can be achieved.

Figure 20 presents the effect of L_am_ on ER under fixed LVF_2_ of 0.1 and various LVF_1_ (0.02–0.1). Obviously, for LVF_1_ = 0.02, 0.04, and 0.06, it can be seen that the change in ER is relatively evident, while for LVF_1_ = 0.08 and 0.1, the change in ER is pretty small. The results are similar to the variation of the optimal L_am_ with LVF in Figure 16, Figure 18 and Figure 19, namely the changing trend of optimal L_am_ is irregular.

In addition, the results of optimal L_am_ and ER under other LVF are presented in Table 5 and Table 6.

With the results of Section 3.2, it can be found that the relation of optimal L_am_ with LVF_1_ and LVF_2_ is irregular since optimal L_am_ is influenced by optimal L_pm_. When the optimal L_pm_ is more than 8 mm, the optimization of L_am_ is not evident, or the influence of L_am_ is not distinct. Nevertheless, when the optimal L_pm_ is less than 8 mm, the influence of L_am_ will be significant. The optimum L_am_ are in the range of 6–21 mm, and they do not deviate much from the L_am_ of the baseline ejector model (15 mm).

### 3.3. A Combination of Optimal L_pm_ and L_am_

Figure 21 depicts the relation of the sum of optimal L_pm_ and L_am_ (L_pam_) with LVF_2_ under changed LVF_1_. It can be observed that for each LVF_1_, the optimal L_pam_ decreases with LVF_2_. Furthermore, the optimal L_pam_ increases with LVF_1_.

Figure 22 indicates the relation of the sum of optimal L_pm_ and L_am_ with LVF_1_ under different LVF_2_. Generally speaking, the optimal L_pam_ increases with LVF_1_ but reduces with LVF_2_, and the relation between optimal L_pm_ and L_pam_ is regular.

## 4. Conclusions

This paper numerically optimizes two mixing chamber geometries of a two-phase ejector under various primary and secondary inlet LVFs. The most important findings are given below:(1)When the primary inlet of the ejector contains liquid while the secondary inlet does not, the optimal L_pm_ and L_am_ are ranged between 23–44 mm and 15–18 mm. When the secondary inlet contains liquid while primary inlet does not, these two optimal lengths are ranged 2–5 mm and 9–15 mm, while when both the primary inlet and secondary inlet contain liquid, they are in the range of 5–23 mm and 6–18 mm, respectively. Thus, two mixing chamber lengths largely depend on the vapor or liquid state of the two inlets;(2)When primary inlet LVF is fixed and secondary inlet LVF increases from 0 to 0.1, the optimal L_pm_ decreases along with the growth of secondary inlet LVF; when secondary inlet LVF is fixed and primary inlet LVF varies from 0 to 0.1, the optimal L_pm_ increases along with the growth of primary inlet LVF;(3)The sum of optimal L_pm_ and optimal L_am_ increases with the increase of primary inlet LVF but decreases with the increase of secondary inlet LVF.

## Figures and Tables

**Figure 1 entropy-25-00007-f001:**
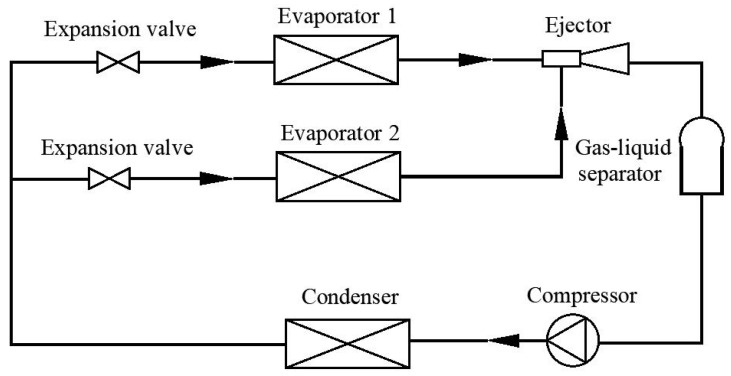
Schematic of a two-temperature evaporator-based EMERS.

**Figure 2 entropy-25-00007-f002:**
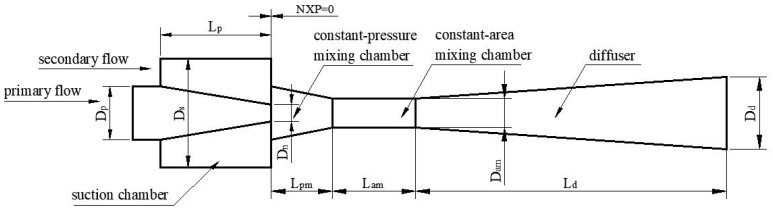
Schematic of the ejector.

**Figure 3 entropy-25-00007-f003:**
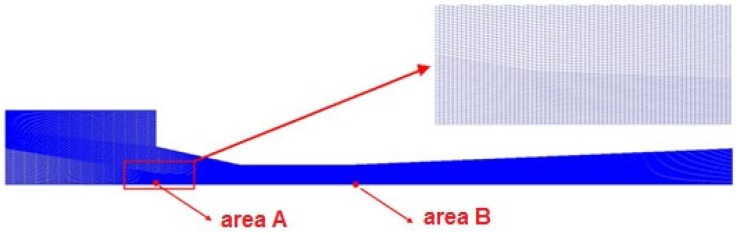
Densified meshes of the ejector.

**Figure 4 entropy-25-00007-f004:**
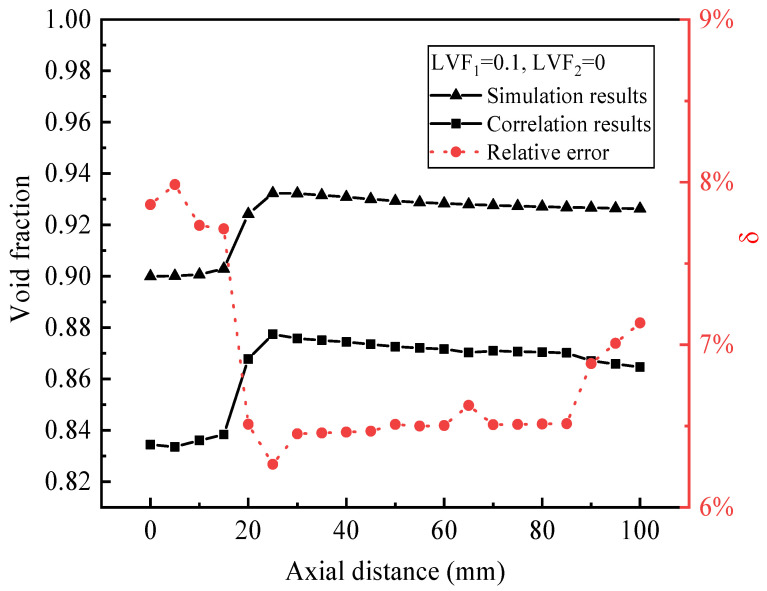
Comparison of α between simulation and correlation results.

**Figure 5 entropy-25-00007-f005:**
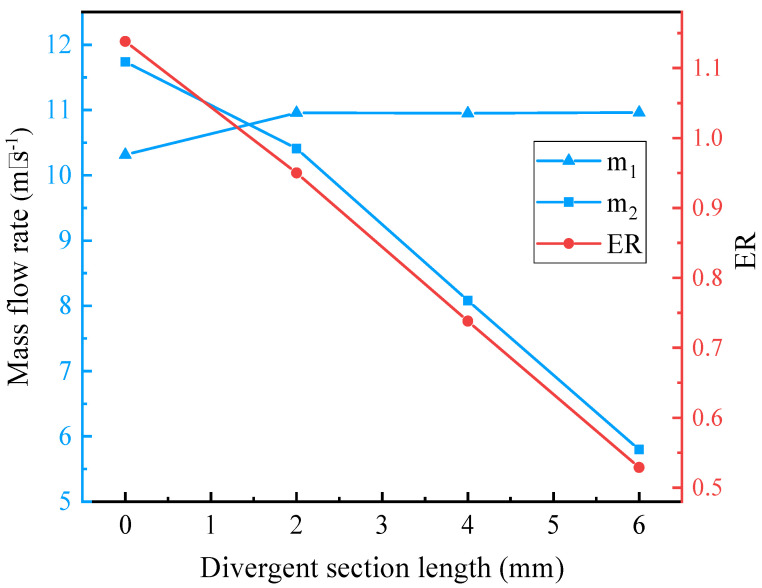
The relation of mass flow rate and ER with diverging section length of the primary nozzle.

**Figure 6 entropy-25-00007-f006:**
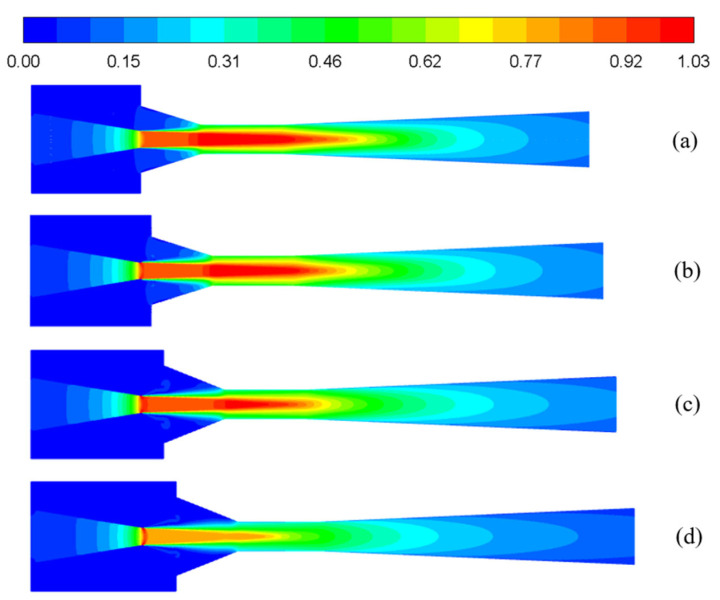
The Mach number contours under different nozzle diverging section lengths: (**a**) 0 mm; (**b**) 2 mm; (**c**) 4 mm; and (**d**) 6 mm.

**Figure 7 entropy-25-00007-f007:**
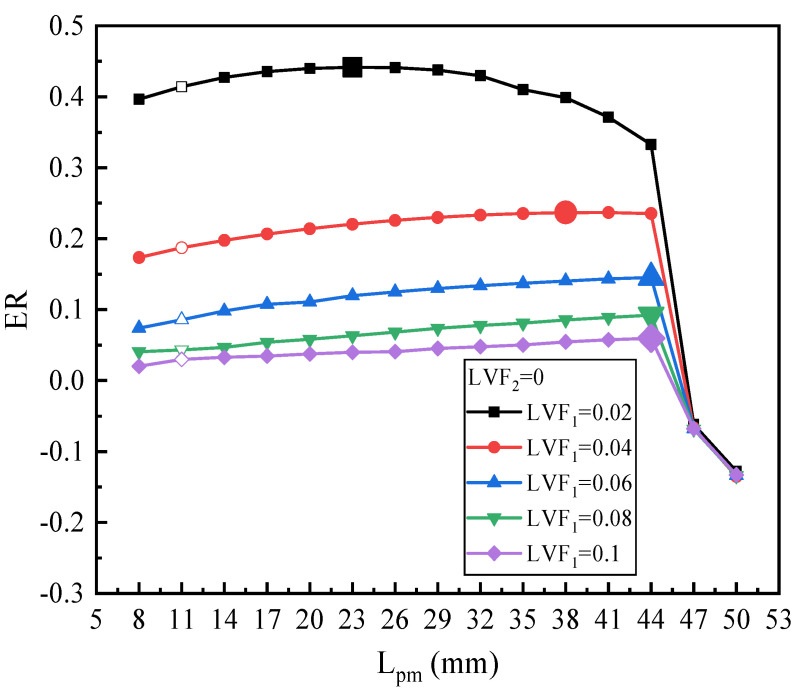
Optimization of L_pm_ on ER under LVF_1_ = 0.02~0.1 (LVF_2_ = 0).

**Figure 8 entropy-25-00007-f008:**
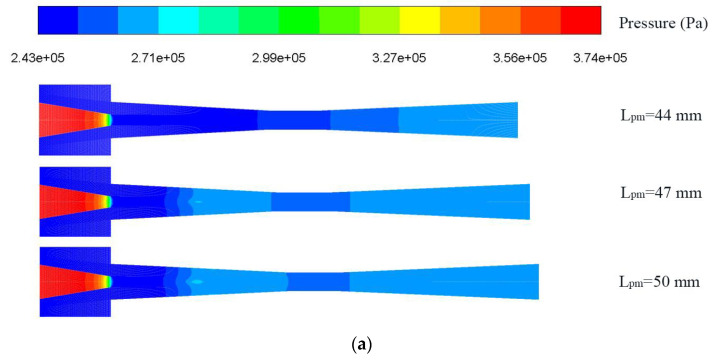
Pressure inside the ejector with different L_pm_ under LVF_1_ = 0.06 and LVF_2_ = 0: (**a**) Contour of static pressure; (**b**) Static pressure along the centerline; (**c**) Contour of velocity.

**Figure 9 entropy-25-00007-f009:**
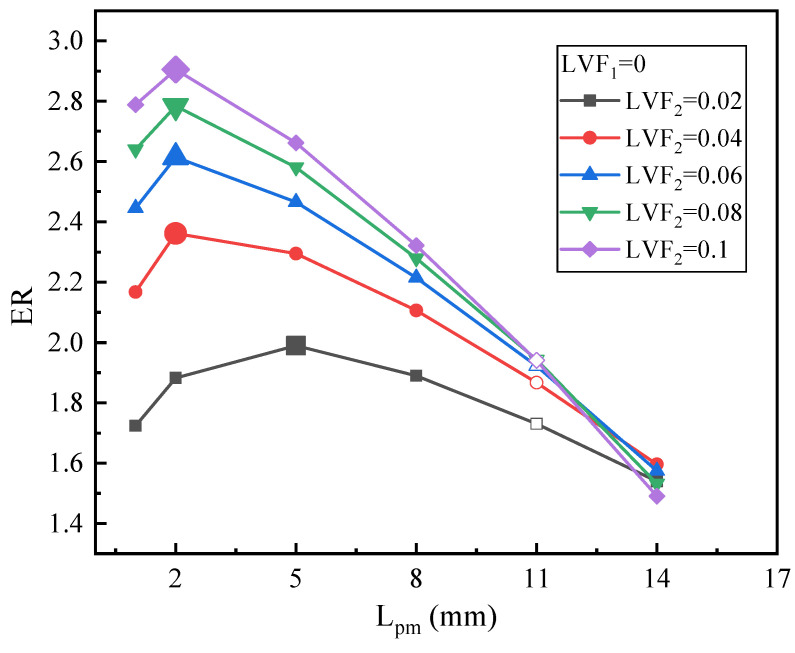
Optimization of L_pm_ on ER with LVF_1_ = 0 and LVF_2_ = 0.02~0.1.

**Figure 10 entropy-25-00007-f010:**
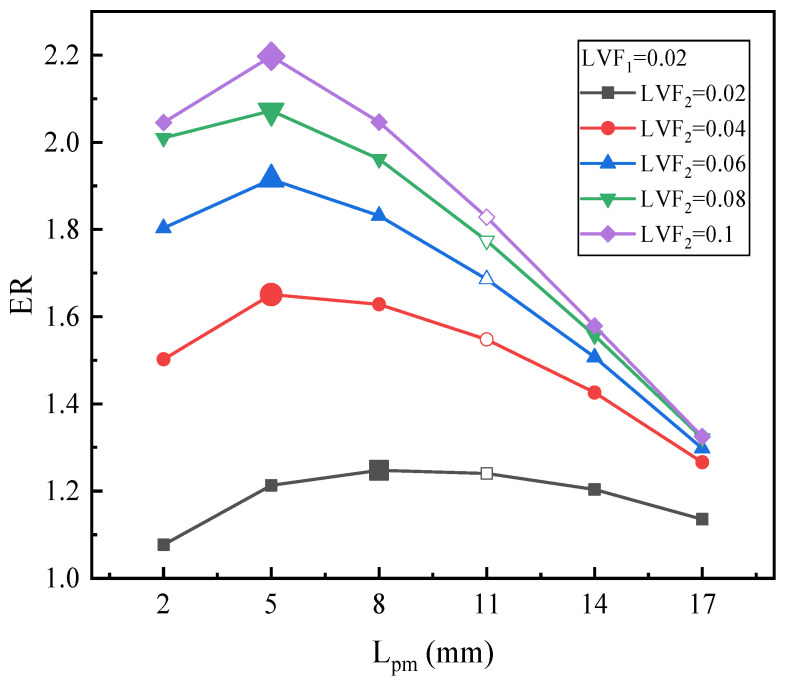
The relation of ER with L_pm_ under LVF_1_ = 0.02 and LVF_2_ = 0.02~0.1.

**Figure 11 entropy-25-00007-f011:**
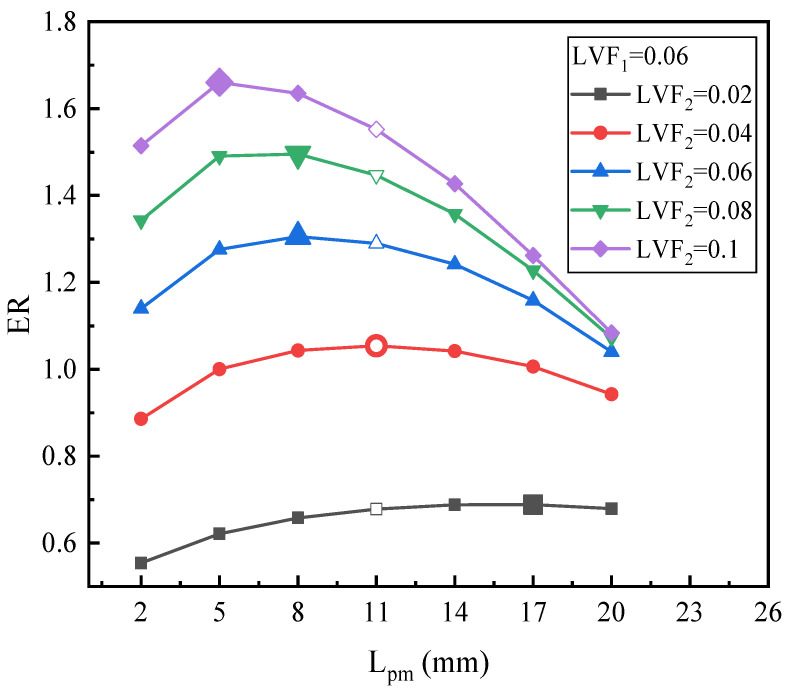
The relation of ER with L_pm_ under LVF_1_ = 0.06 and LVF_2_ = 0.02~0.1.

**Figure 12 entropy-25-00007-f012:**
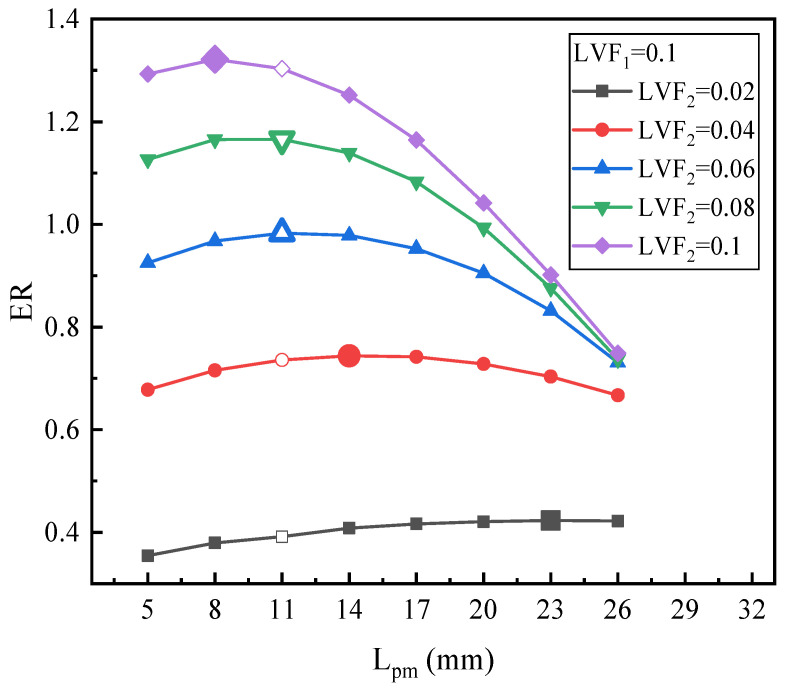
The relation of ER with L_pm_ under LVF_1_ = 0.1 and various LVF_2_.

**Figure 13 entropy-25-00007-f013:**
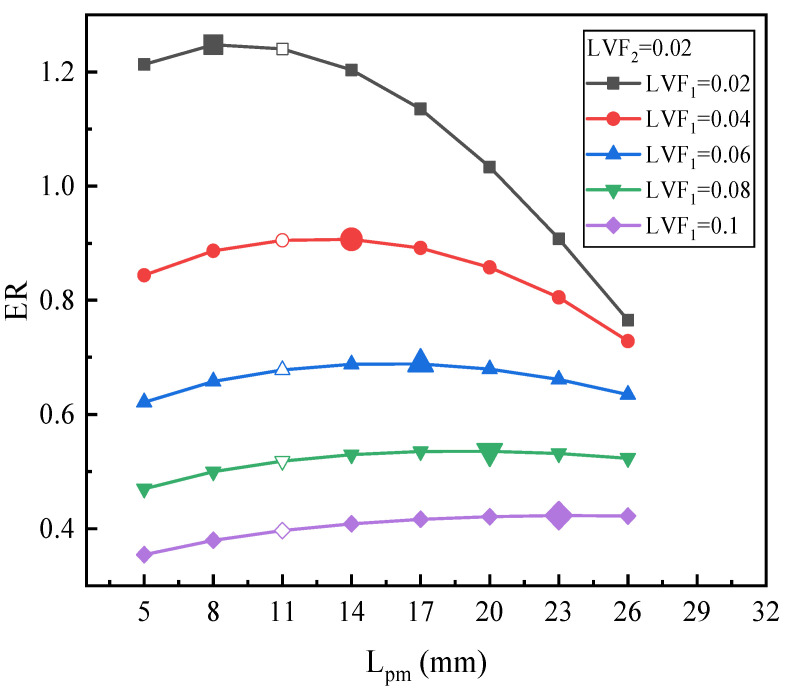
The relation of ER with L_pm_ under LVF_2_ = 0.02 and LVF_1_ = 0.02~0.1.

**Figure 14 entropy-25-00007-f014:**
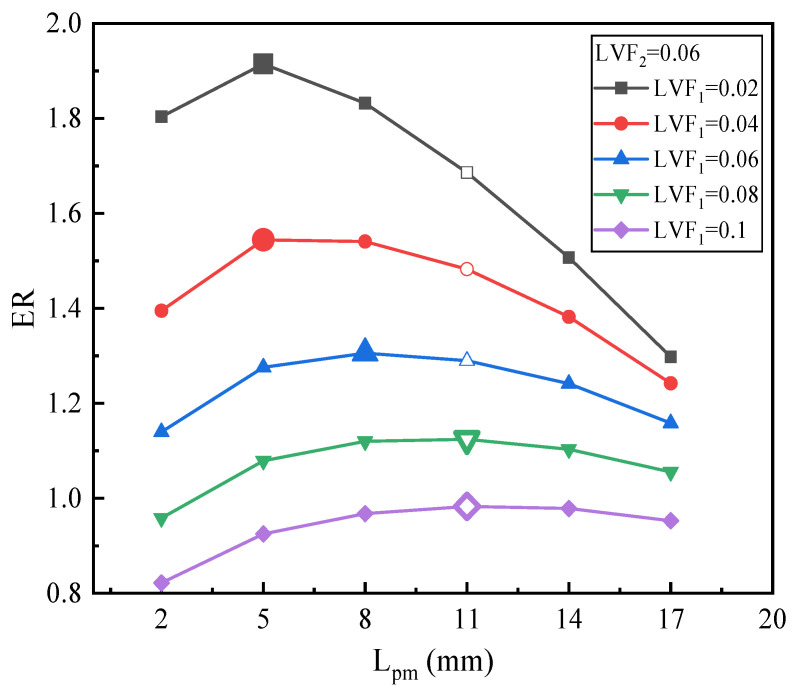
The relation of ER with L_pm_ under LVF_2_ = 0.06 and LVF_1_ = 0.02~0.1.

**Figure 15 entropy-25-00007-f015:**
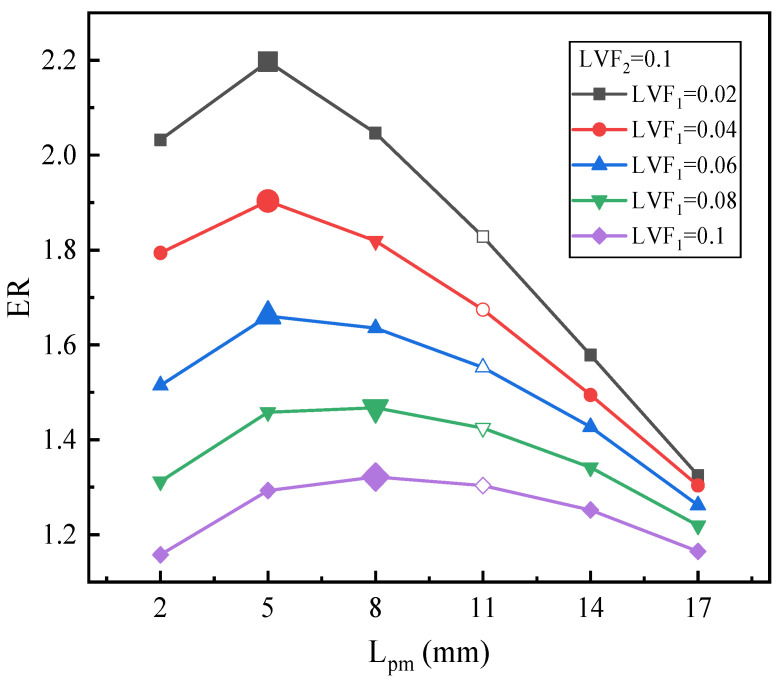
The variation of ER with L_pm_ under LVF_2_ = 0.1 and LVF_1_ = 0.02~0.1.

**Figure 16 entropy-25-00007-f016:**
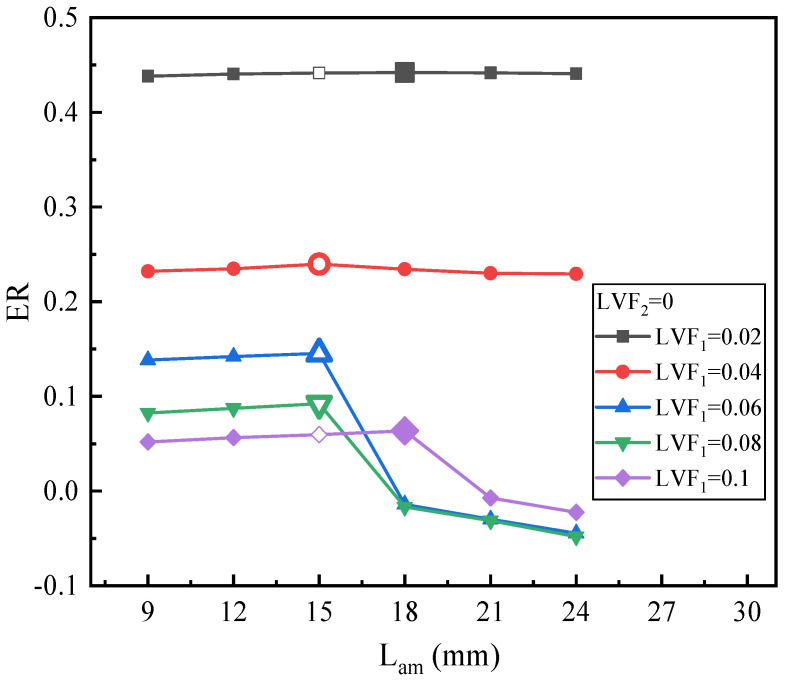
ER with L_am_ under LVF_1_ = 0.02~0.1 (LVF_2_ = 0).

**Figure 17 entropy-25-00007-f017:**
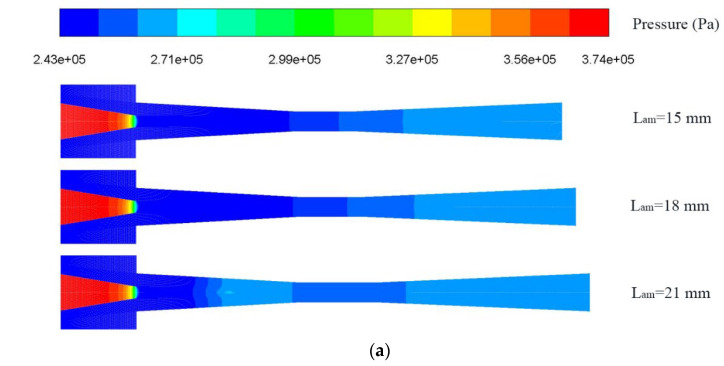
Pressure inside the ejector with different L_am_ under LVF_1_ = 0.1 and LVF_2_ = 0: (**a**) Contour of static pressure; (**b**) Static pressure along the centerline; (**c**) Contour of velocity and (**d**) Velocity vector field of L_am_ = 21 mm.

**Figure 18 entropy-25-00007-f018:**
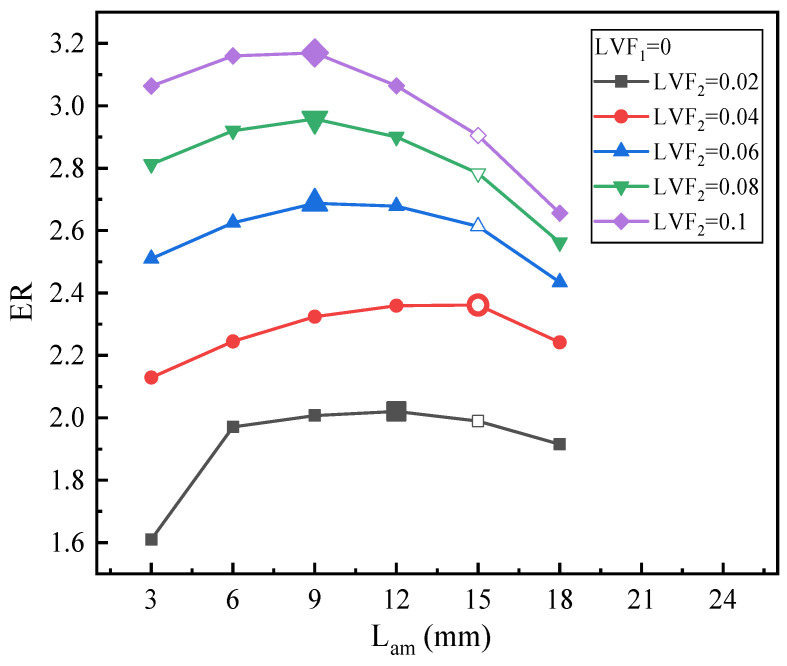
ER with L_am_ under LVF_2_ = 0.02~0.1 (LVF_1_ = 0).

**Figure 19 entropy-25-00007-f019:**
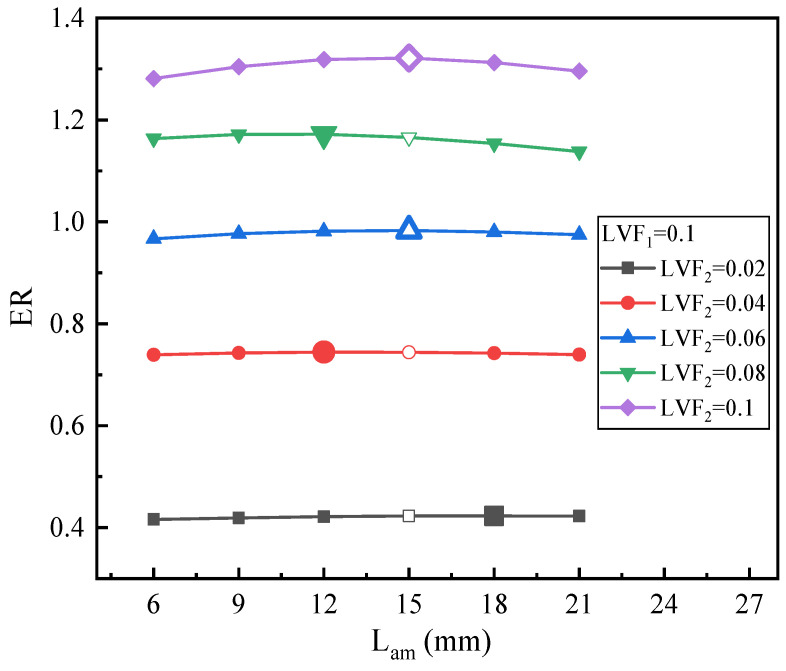
The relation of ER with L_am_ under LVF_1_ = 0.1 and LVF_2_ = 0.02~0.1.

**Figure 20 entropy-25-00007-f020:**
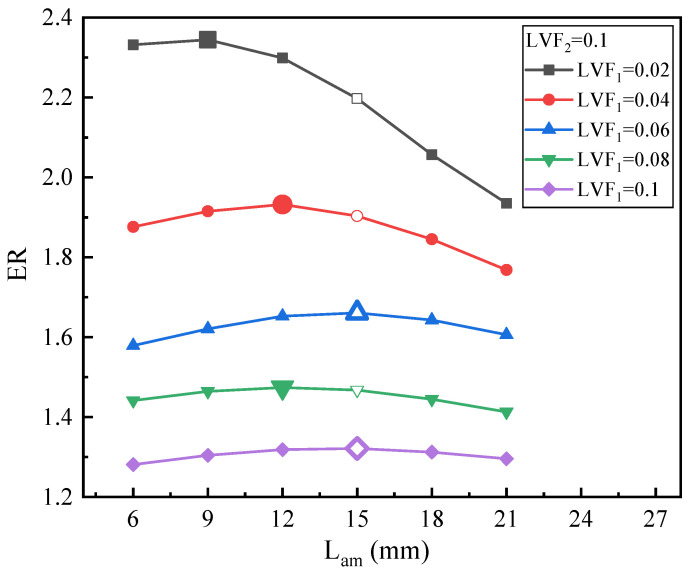
The relation of ER with L_am_ under LVF_2_ = 0.1 and LVF_1_ = 0.02~0.1.

**Figure 21 entropy-25-00007-f021:**
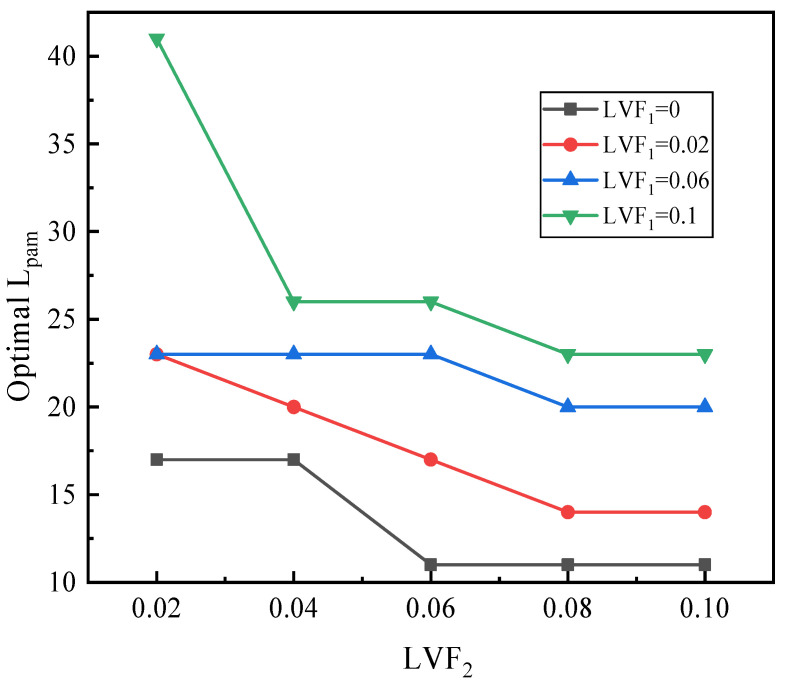
The relation of optimal L_pam_ with LVF_2_ under different LVF_1_.

**Figure 22 entropy-25-00007-f022:**
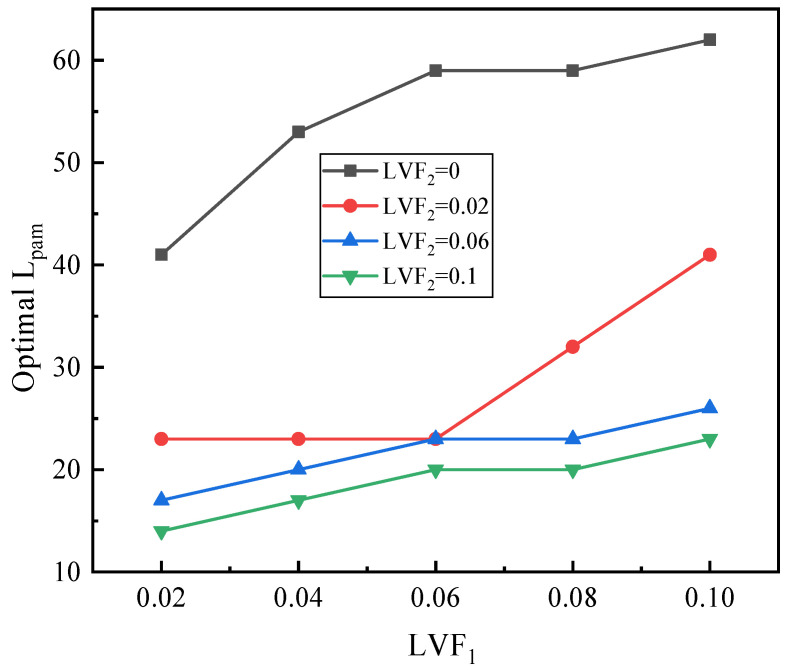
The relation of optimal L_pam_ with LVF_1_ under given LVF_2_.

**Table 1 entropy-25-00007-t001:** Initial geometrical parameters of the ejector.

Parameters	Value (mm)
The suction chamber diameter, D_s_	19.6
The primary nozzle inlet diameter, D_p_	9.6
The primary nozzle outlet diameter, D_n_	3
The constant-area mixing chamber diameter, D_am_	5.2
The diffuser outlet diameter, D_d_	10
The primary nozzle length, L_p_	20
The constant-pressure mixing chamber length, L_pm_	11
The constant-area mixing chamber length, L_am_	15
The diffuser length, L_d_	56

**Table 2 entropy-25-00007-t002:** Boundary conditions for the ejector.

Parameters	Type	Pressure (kPa)	Temperature (K)	
			Superheated Gas	Two-Phase
Primary inlet	Pressure inlet	374.6	290	280
Secondary inlet	Pressure inlet	243.3	278	268
Outlet	Pressure outlet	267.63	-	-

**Table 3 entropy-25-00007-t003:** Mesh sensitivity analysis of area A.

Grid Number	Pressure (Pa)	Error (%)	Velocity (m·s^−1^)	Error (%)
52,200	274,947	-	120.138	-
68,520	275,288	0.12	119.893	0.2
83,100	275,521	0.085	119.684	0.17
114,880	275,613	0.033	119.623	0.051

**Table 4 entropy-25-00007-t004:** Mesh sensitivity analysis of area B.

Grid Number	Pressure (Pa)	Error (%)	Velocity (m·s^−1^)	Error (%)
52,200	206,538	-	165.218	-
68,520	206,879	0.165	164.934	0.172
83,100	207,054	0.085	164.841	0.056
114,880	207,123	0.033	164.788	0.032

**Table 5 entropy-25-00007-t005:** Results of optimal L_am_ and maximum ER under varied LVF_1_ for a given LVF_2_.

	LVF_2_ = 0 (ER)	LVF_2_ = 0.02 (ER)	LVF_2_ = 0.06 (ER)	LVF_2_ = 0.1 (ER)
LVF_1_ = 0.02	18 (0.442)	15 (1.248)	12 (1.941)	9 (2.344)
LVF_1_ = 0.04	21 (0.24)	9 (0.912)	15 (1.544)	12 (1.932)
LVF_1_ = 0.06	15 (0.145)	6 (0.693)	15 (1.306)	15 (1.661)
LVF_1_ = 0.08	15 (0.092)	12 (0.536)	12 (1.128)	12 (1.474)
LVF_1_ = 0.1	18 (0.065)	18 (0.423)	15 (0.983)	15 (1.321)

**Table 6 entropy-25-00007-t006:** Results of optimal L_am_ and maximum ER under varied LVF_2_ for a given LVF_1_.

	LVF_1_ = 0 (ER)	LVF_1_ = 0.02 (ER)	LVF_1_ = 0.06 (ER)	LVF_1_ = 0.1 (ER)
LVF_2_ = 0.02	12 (2.02)	15 (1.248)	6 (0.693)	18 (0.423)
LVF_2_ = 0.04	15 (2.362)	15 (1.651)	12 (1.055)	12 (0.744)
LVF_2_ = 0.06	9 (2.687)	12 (1.941)	15 (1.306)	15 (0.983)
LVF_2_ = 0.08	9 (2.958)	9 (2.158)	12 (1.504)	12 (1.172)
LVF_2_ = 0.1	9 (3.17)	9 (2.344)	15 (1.661)	15 (1.321)

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
