# Peer review of "Optimization of Two-Phase Ejector Mixing Chamber Length under Varied Liquid Volume Fraction"

_entropy, 2022, doi:10.3390/e25010007_

Round 1

Reviewer 1 Report

  This paper investigates the influence of mixing chamber on ejector performance through numerical method. Linguistics, readability and structure of the paper is well. However, there are some concerns or questions which should be addressed by the authors:    

1.     Figures need to be consistent in size and font size. Figures caption need to have more details.

2.     For convenience of reference, the boundary conditions should be given in this paper. Authors could list them in a Table.

3.     The geometry of constant-pressure mixing section, constant-area mixing section and nozzle should be given in the manuscript.

4.     In Figure 5, what is the reason that the flow rate increases first and then remains unchanged with the divergent section?

5.     According to Figures 8(a) and 8(c), as we know, the fluid pressure is related to velocity. However, the distribution of pressure contour and velocity contour is different. Please explain that. 

6.     The optimal of Lpm corresponding the LVF1 and LVF2 are different, the influence of Lpm on ER with LVF1 and LVF2 should be studied in same range.

Reviewer 2 Report

Additional request for comparative analysis of the CFD modeling results of this study and the data of actual experiments.

Please explain in detail about the liquid volume fraction (LVF) in the menu script.

Explain the words ER, PRR, m1, m2, etc. in the menu script.

The angle along with the length of Lpm in this study is an important influencing variable. Is the angle constant? Angle optimization is required. In addition, the angle of the diffuser part should also be considered.

In Table 1 and Fig. 3, the average analysis of the cross-sectional areas of A and B seems more objective than the pressure and velocity analysis of points A and B.

In Figure 16, explain why the result value of the LVF1=0.1 condition is different.

In this study, Lpm and Lam were separated and optimized. Optimize for Lpm and Lam simultaneously

Reviewer 3 Report

The present paper explores the performance of the ejector under different inlet fluid conditions through CFD technology, which is a good attempt. However, I believe that the article should not published in the present from due to the following comments:

-1. The Abstract should be rewritten. It should contain answers to the following questions: What problem was studied and why is it important? What is the novelty of the work and where does it go beyond previous efforts in the literature? What methods were used? What are the important results? What key conclusions can be drawn from the results? Include specific and quantitative results in the Abstract, while ensuring that it is suitable for a broad audience.

-2. As a complete paper, the CFD model should be described.

-3. The geometry of initial ejector should be given.

-4. References for CFD model validation should be listed

-5. The operating conditions of the ejectors should be given in the Section 3.1.

-6. During the optimization process, other structural dimensions of the ejector should be given, such as the angle of the constant-pressure mixing chamber and the diameter of the mixing chamber.

-7. In previous studies, the influence of mixing chamber diameter is usually enormous, whether the author should consider the influence of mixing chamber diameter.

-8. Conclusions should be improved.

-9. There many technical and editorial mistakes in the manuscript.

-10. Adding a Nomenclature could be helpful to improve the manuscript.

-11. There is no any entropy graph, which is the main objective of Entropy journal.

-12. English needs to be improved.

Thus, with the previous comments I believe the authors might improve the paper and provide a different approach in which the novelty is better shown.

Round 2

Reviewer 1 Report

accept

Author Response

Thanks for your approval of this manuscript with "accept".

Reviewer 2 Report

Thank you for responding

1. Additional request for comparative analysis of the CFD modeling results of this study and the data of actual experiments.

Response: Thank you very much for your valuable suggestions. In fact, the void fraction used to validate the CFD are the correlations of experimental results.

=> Reviewer's response : My comment is to verify the CFD modeling results based on the actual experimental condition results including actual porosity.

 4. The angle along with the length of Lpm in this study is an important influencing variable. Is the angle constant? Angle optimization is required. In addition, the angle of the diffuser part should also be considered.

Response: Thank you very much for your valuable suggestions. In this study, the angle of constant-pressure section is not constant, while the angel of diffuser part is constant.

=> Reviewer's response : If the angle of the Lpm part is not constant, is it possible to compare the CFD modeling results for each condition?

5. In Table 1 and Fig. 3, the average analysis of the cross-sectional areas of A and B seems more objective than the pressure and velocity analysis of points A and B.

Response: Thank you very much for your valuable suggestions. The cross-sectional areas of A and B are constant.

=> Reviewer's response : It is necessary to analyze with Mesh sensitivity analysis of area A and B rather than Mesh sensitivity analysis of Point A and B.
